# Crashworthiness Study of 3D Printed Lattice Reinforced Thin-Walled Tube Hybrid Structures

**DOI:** 10.3390/ma16051871

**Published:** 2023-02-24

**Authors:** Chenglin Tao, Xin Zhou, Zeliang Liu, Xi Liang, Wentao Zhou, Huijian Li

**Affiliations:** 1School of Civil Engineering and Mechanics, Yanshan University, Qinhuangdao 066004, China; 2Hebei Key Laboratory of Mechanical Reliability for Heavy Equipments and Large Structures, Yanshan University, Qinhuangdao 066004, China

**Keywords:** crashworthiness, 3D printing, material reinforcement, energy absorption, gradient structure

## Abstract

Based on the advantages of thin-walled tubes and lattice structures in energy absorption and improved crashworthiness, a hybrid structure of lattice-reinforced thin-walled tubes with different cross-sectional cell numbers and gradient densities was constructed, and a high crashworthiness absorber with adjustable energy absorption was proposed. The experimental and finite element characterization of the impact resistance of uniform density and gradient density hybrid tubes with different lattice arrangements to withstand axial compression was carried out to investigate the interaction mechanism between the lattice packing and the metal shell, and the energy absorption of the hybrid structure was increased by 43.40% relative to the sum of its individual components. The effect of transverse cell number configuration and gradient configuration on the impact resistance of the hybrid structure was investigated, and the results showed that the hybrid structure showed higher energy absorption than the empty tube, and the best specific energy absorption was increased by 83.02%; the transverse cell number configuration had a greater effect on the specific energy absorption of the hybrid structure with uniform density, and the maximum specific energy absorption of the hybrid structure with different configurations was increased by 48.21%. The gradient density configuration had a significant effect on the peak crushing force of the gradient structure. In addition, the effects of wall thickness, density and gradient configuration on energy absorption were quantitatively analyzed. This study provides a new idea to optimize the impact resistance of lattice-structure-filled thin-walled square tube hybrid structures under compressive loading through a combination of experiments and numerical simulations.

## 1. Introduction

In recent years, with the growth of the transportation industry and the increased speed requirements of vehicles, the passive safety of vehicles has become increasingly important by introducing more efficient systems for vehicles that maintain the structural integrity of the vehicle and enhance passenger safety in a crash [1,2]. At the same time, growing concerns about environmental pollution and sustainability are driving lightweight design to reduce the fuel consumption of vehicles. As an effective energy absorbing element with a simple structure and controlled and ordered deformation, thin-walled structures absorb energy through plastic deformation and convert kinetic energy into internal energy to reduce the load when subjected to impact loads [3,4]. A large number of studies have been conducted to analyze and optimize the energy absorption performance of thin-walled structures using experimental, numerical simulation, and theoretical prediction [5,6,7] methods. The compression properties of these structures were enhanced by various materials, cross-sectional shapes and geometric patterns, and the results showed that the cross-sectional shape design and lightweight material filling had the most significant effect on the energy absorption properties [8,9,10].

Qiu et al. [11] comprehensively investigated the collision behavior of different multi-cell element hexagonal section columns under axial and oblique loading. It was found that the number of angles played an important role in improving the energy absorption for the same number of cytons. In addition, Sun et al. [12] conducted experimental and numerical simulations of the collisional properties of cross-shaped tubes. The results showed that the energy absorption of cross-shaped tubes was 2.5 times higher than that of square tubes. Zhang et al. [12,13] studied the compression behavior of multi-cell structures, and the specific energy absorption (SEA) could be increased by 78.25% when the cross section of a single structure was transformed into a multi-cell element structure. However, the growth trend of energy absorption decreased as the number of cells increased. Sun et al. first performed collision analysis and the multi-objective optimization of thin-walled structures with axial gradient thickness [12,14], and the specific energy absorption could be improved by 45.6% when the peak force was controlled at the same level. Pangtong et al. [15] proposed a multi-cell tube with axially varying thickness, and this structure not only reduced 20% of the initial peak force, it also maintained the energy absorption capacity for relatively long compression displacements. However, the thin-walled tube is very sensitive to preparation defects. These defects tend to lead to instability under axial compressive loading, and the load-bearing capacity and energy-absorbing performance need to be further improved.

Another important and effective way to improve the energy absorption capacity of thin-walled structures is the addition of lightweight fillers. Porous materials represented by foam metals exhibit good impact resistance [16,17], with relatively low density and relatively stable stresses [18]. Metal-based composite foams can exhibit a long and slowly rising platform, ensuring a large energy absorption capacity, and their properties can be tailored over a wide range. In addition, the filler causes a shift in the deformation mode and a positive interaction between the tube wall and the filling material. Goel et al. [19] investigated the deformation, energy absorption, and compression behavior of square and round tubes with single, double, and multi-wall foam fillers. It was shown that thin-walled tubes with foam fillers had larger deformation, while improving over 30% over energy absorption. Hanssen et al. [4,20] presented approximate formulas to predict the response of foam-filled aluminum tubes under quasi-static and dynamic loading conditions. Their study showed that the total absorbed energy of the foam-filled tube exceeded the sum of the energy absorbed by the empty column and the foam, due to the interaction effect. Similarly, Fang et al. [21,22,23] investigated the dynamic mechanical properties of thin-walled structures filled with functionally graded foam and showed that functionally graded foam structures absorbed more than 15% more energy than uniform foam of the same mass. Despite the progress in the design of efficient thin-walled tubes by adding new structures and materials, some inherent potential defects, such as cracks, fractures, inclusions, or discontinuities, can significantly affect the energy absorption of the tubes.

With the development of additive manufacturing techniques, greater freedom is provided for structural design, allowing the production of structures with more complex geometric features as new filler materials. One of them is lattice structures with good energy absorption [18], which becomes another ideal filler material to improve the impact resistance of thin-walled tubes. Beharic et al. [24] used additive manufacturing techniques to fabricate tensile-dominated and bending-dominated lattice structures. The mechanical properties, failure mechanisms, and energy absorption characteristics of the BCC structures were systematically investigated through theoretical analysis, numerical simulations, and experiments. The results showed that the bending-dominated BCC lattice structure had low strength and stiffness, but good energy absorption capability, and its failure exhibited obvious shear bands. Bai et al. [25] formed a series of lattice structure metamaterials with different densities and configurations by varying the supporting rod angle of uniform lattice cell elements and then assembling them into size-graded lattice superstructures with different densities and topological configurations. The influence of the compression response and its gradient orientation on the structural properties was investigated by quasi-static compression experiments and numerical simulation results testing.

Cetin et al. [26,27] mixed lattice structures with thin-walled tubes for the first time and investigated the energy absorption behavior of different lattice hybrid structures under axial impact loads using experiments and numerical simulations, and the results showed that the lattice structures helped to improve the flexural and bending resistance of thin-walled tubes, and the impact energy of the hybrid structures was increased by 115% compared to the sum of individual components. Li et al. [28] proposed a complex scale evaluation method and discrete optimization algorithm to design and optimize the topology of thin-walled tubes filled with lattice structures, thus further confirming the effect of different topological lattice materials on the weight lightening and high strengthening of hybrid structures. By combining porous square tubes with lattice metamaterials, Liu et al. [29] investigated the effect of lattice material fillers on the overall energy absorption and deformation patterns of the structure. The results showed that the multi-chambered tube filling showed a significant improvement in energy absorption and compressive force efficiency compared to a single tube, and the energy absorption performance of the hybrid structure could be significantly improved by the proper selection of design parameters. However, to our knowledge, the effects of lattice structure arrangement, density distribution, porosity, etc. on the energy absorption properties of hybrid structures have rarely been considered.

In this paper, lattice structures with different pore and gradient densities were created by varying the geometric parameters of the BCC cytosol, thus enabling customizable tuning of the mechanical properties. Two types of hybrid structures with uniform densities and graded gradient densities of lattice-structure-filled thin-walled aluminum tubes were proposed, and the local density changes in the lattice structure were achieved by varying the radius of the lattice structure cell support rods, the support rod clamping angle and the number of transverse layer cells. Firstly, quasi-static tests were carried out on the hybrid structure, followed by a numerical study to simulate the tests. This was followed by an in-depth analysis of the effect of design parameters on the impact resistance of the hybrid structure to thus investigate the effect of wall thickness, density, and gradient configuration on the energy absorption of the hybrid structure.

## 2. Materials and Methods

Both lattice structures and thin-walled tubes are widely studied and applied as energy absorbers in engineering and practice. In this study, a hybrid structure using a body-centered cubic (BCC) lattice structure to fill a thin-walled square tube was constructed, and a novel energy absorption device was proposed, as shown in Figure 1. BCC lattice structures with different pores, different morphologies, and different gradient density configurations were created by varying the geometric parameters of the cell elements to achieve diverse mechanical responses and a wide range of mechanical properties, as well as customizable mechanical property tuning. In this section, several lattices with the same relative density but different structural arrangement forms were designed to fill thin-walled aluminum tubes, and the fabrication processes of EDM wire-cutting technique and selective laser melting (SLM) 3D printing used to fabricate hybrid structures with high strength and high specific energy absorption are described. Meanwhile, the printed lattice structure samples were analyzed micro-morphologically using electron microscopy.

### 2.1. Structure Description

In this study, the axial compression of two types of hybrid structures, uniform-density-lattice-filled thin-walled aluminum tubes (ULT) and functional-gradient-density-lattice (GLT)-filled thin-walled aluminum tubes, was considered. The overall size and relative density of the structure was maintained constant by varying the lattice structure cell geometry parameters (including the number of transverse cells m, the number of longitudinal layers n, the cell size, the angle between supporting rods, and the supporting rod radius). The effect of the number of cross-sectional units on the crashworthiness of the uniform density hybrid structures and the effect of gradient configuration on the crashworthiness of gradient density hybrid structures were investigated. The loading conditions were simulated for an axial load on an energy absorbing box of a passenger car vehicle with the bottom end of the tube placed on a fixed bottom surface and the top end subjected to axial compression by a rigid wall, as shown in Figure 2. All thin-walled tubes were specified to have a thickness of 1 mm, an inner wall width of 30 mm and a height of 120 mm, with a design mass of 54.61 g for all of the above hybrid structures.

The ULT hybrid structure lattice was characterized by the fact that all lattice cytons within the same structure had the same envelope size and supporting rod radius, and thus the structure had a uniform density. Three BCC lattice configurations with different structural arrangements were considered by adjusting the number of transverse cytons m and the number of longitudinal layers n. Figure 3a–c show the lattice structures with different numbers of cytons in the cross-sectional length direction. The three configurations include first-, second-, and third-order cytons arranged to form three lattice structures with three sizes of pores, in which the numbers of cytons are 1 × 1 × 4, 2 × 2 × 8, 3 × 3 × 12. The supporting rod diameters are 3 mm, 1.5 mm, and 1 mm, respectively, and are named ULT1, ULT2, and ULT3 according to the number of cross-sectional lattice cells.

The lattice of the GLT hybrid structure had different density distributions along the axial direction, which was characterized by the same supporting rod diameter for each layer of the lattice structure, and two different gradient arrangements of the lattice structure were considered by changing the angle between the lattice supporting rods and the supporting rod diameter, with the same number of lattice cytons as ULT2. As shown in Figure 3d for the variable angle gradient lattice hybrid tube (VGLT), the gradient variation of the lattice structure is caused by the change of the supporting rod angle with different cell height in different layers, the same layer cell has the same configuration, the cell height varies from 8 mm, 10 mm, 12 mm, 14 mm, 16 mm, 18 mm, 20 mm to 22 mm, and the supporting rod diameter of the single cell is 1.5 mm. Figure 3e shows the layered-gradient-lattice-filled thin-walled tube hybrid structure (LGLT), where the gradient variation of the structure is caused by changing the supporting rod radius of different layers, while the supporting rod radius of the same lattice layer is the same, and the relative density of the structure decreases from bottom to top as 0.093, 0.083, 0.073, 0.063, 0.053, 0.043, 0.033 and 0.023.

### 2.2. Materials and Manufacturing

The square tubes were machined by electrical discharge machine (EDM) wire-cutting from a highly ductile aluminum alloy (Al-1060), which is a lightweight alloy with high strength. Thin-walled square aluminum tube specimens were made by cutting aluminum alloy blocks using the EDM wire-cutting method. The aluminum block was first fixed in the working area, then the machining route was drawn and the process was started by the control system. During the machining process, a cooling fluid was continuously injected to prevent the molybdenum wire from melting due to the high temperatures.

Selective laser melting (SLM) was chosen for the fabrication of BCC lattice structures for filled thin-walled tubes due to its excellent forming efficiency. In order to obtain a lattice structure with high specific strength and high specific energy absorption, the AlSi10Mg material was chosen for the fabrication of the samples, and its chemical composition is listed in Table 1. The SLM process was carried out in an EOS 280 3D printer equipped with a 400 W Yb fiber laser with a laser beam diameter of approx. 100 μm. The laser beam rapidly melted the AlSi10Mg aluminum alloy powder by thermal action and then gradually solidified into solid layers which were stacked layer by layer. Process parameters, such as laser exposure, scanning speed, layer thickness and filling distance, were set according to the manufacturer’s recommendations to achieve a fully dense product (approx. 99.8%). In this work, all samples were fabricated along the same orientation to eliminate the effect of print orientation. To prevent oxidation during the SLM process, all samples were fabricated in an argon atmosphere with oxygen concentrations controlled to below 100 ppm. To make the structure and composition of AlSi10Mg more homogeneous, to relieve the thermal stresses generated by the high cooling rate during the SLM process, and to improve the ductility of the parts, the samples were heat treated in a vacuum furnace, where all samples were heated to 500 °C at 200 °C/h and held at this temperature for 2 h before being cooled naturally to room temperature in air. Finally, they were sandblasted with overpressure air to remove particles adhering to the surface.

To obtain the true stress–strain curves of the materials, dog bone tensile samples were fabricated using the same materials as the specimens (i.e., Al-1060 and AlSi10Mg) according to the ASTM E8/E8M standard with the same wire-cutting and SLM process parameters to plot the stress–strain curves of the materials by uniaxial tensile tests. A Zwick Z010 universal testing machine with 100 kN capacity was used to perform standard tensile tests to obtain the mechanical properties of the materials, and the loading rate was kept constant at 2 mm/min. The true stress–strain curves and tensile samples for both materials and the test setup are shown in Figure 4, while the mechanical properties are shown in Table 2.

Figure 5a shows the macroscopic morphology of the lattice samples with different diameter dimensions. The supporting rods were well formed with no obvious holes, cracks, missing or other defects, which indicated that the process parameters were suitable for fabricating BCC lattice structures. To further evaluate the quality of the processed samples, scanning electron microscopy (SEM) was used to examine the surface morphology, as shown in Figure 5b. The non-melted powder adhering to the sample surface can be observed, and the supporting rod surface was rough. One reason was the incomplete melting and binding of the powder particles due to insufficient laser energy and the high cooling rate, which may have led to poor fusion between adjacent molten powder channels. In addition, residual gases in the melt pool can create small holes in the sample after solidification.

The weight measurements of the printed samples are shown in Table 3. As can be seen from the table comparing the weights of the experimental and design samples, the measured weights of the printed samples were all slightly larger than the design values of their CAD models, and the error increased as the column diameter of the lattice structure decreased, with a maximum error of 1.55 g. In addition, the Young’s modulus obtained from the quasi-static tensile test was 63.70 GPa, which was smaller than the normal value for aluminum alloys of 70 GPa, and the smaller Young’s modulus can be attributed to the presence of porosity in the 3D-printed samples. In conclusion, the deviation between the actual porosity and the theoretical porosity was caused by the instability of the melt flow and powder adhesion due to the SLM process-specific limitations, which was an unavoidable deviation between the sample design and the actual experiment.

## 3. Experimental Setup and Finite Element Model

In order to study the mechanical response and deformation mechanism of the hybrid structure under axial loading conditions, quasi-static axial compression tests and finite element analysis were performed on the hybrid structure in this section, and various crashworthiness evaluation indices were defined.

### 3.1. Quasi-Static Compression Experiments

In order to investigate the mechanical response and deformation mechanism of the hybrid structure, this paper performed compression tests on the hybrid structure under quasi-static compression loading conditions, which were performed at a compression rate of 2 mm/min on an electronic universal testing machine. In the compression test, the specimen was placed between two platens without clamping, and the sampling rate was set to 50 Hz. The force and displacement results were recorded by an automatic data acquisition system, and the deformation history of the specimen was obtained by recording the deformation pattern of the specimen during compression by a camera. The compression displacement was 90 mm, which was 75% of the original total height, and the experimental setup is shown in Figure 6. All experiments were repeated three times and averaged for mapping analysis to ensure the reliability of the results obtained for all structures under quasi-static loading conditions. Finally, the impact resistance and energy absorption capacity of the hybrid tube were evaluated based on the failure strain behavior and force–displacement response curves obtained from the experiments.

### 3.2. Finite Element Analysis Models

In this work, numerical simulations of the structure were performed using Ls-Dyna, which is a double-accuracy nonlinear finite element display software, to study the crashworthiness of the hybrid structure under axial loads. The finite element model of the hybrid structure ULT1, which is representative of all the models, is shown in Figure 7. The model consisted of a rigid upper and lower support platform and specimens. The rigid plates at the top and bottom of the model were used to simulate the upper and lower compression plates in the experiment. An axial displacement load was applied to the upper plate, and all degrees of freedom of the lower plate were constrained. In order to make the numerical simulation consistent with the experimental deformation pattern, an initial defect was introduced as a trigger, i.e., the node was shifted inward by 0.1 mm at a half a fold half-wavelength from the top. Due to the small depth and area of the trigger, it was only used to control the location where the first fold appeared in the axial load and had no significant effect on the energy absorption.

Thin-walled tubes and rigid plates were discretized using a Belytschko–Tsay simplified shell cell (S4R) with four nodes. The BT algorithm uses single-point integration with the advantage of fast computation speed and five integration points in the thickness direction to capture more deformation behavior during axial compression deformation. Such a cell is suitable for simulating large structural deformations and has been used in previous finite element studies. The lattice structure was meshed using linear tetrahedral cells (C3D4), which are solid cells are defined by *SECTION_SOLID. An integration scheme of “constant stress solid cells (default cell type)” was used, stiffness-based hourglass control was used to avoid spurious zero energy deformation modes, and reduced integration was used to avoid volume locking. A total of 450,000 units were calculated using four and a half hours. To account for the computational cost of FEA, the indenter speed was increased to 1 m/s in the first 20 ms and then remained constant for the remainder of the process. The tube is usually compressed under quasi-static conditions to evaluate the energy absorption performance. The loading rate is usually set to 2 mm/min in real quasi-static experiments. However, this is too slow for the explicit finite element algorithm. A lot of time is spent on computation. For numerical simulations, the loading rate was set to 1000 mm/s. This improves the computational efficiency of the finite element model. A large amount of literature shows that the effect of the strain rate of the aluminum alloy material can be neglected when the compression rate is less than 1000 mm/s. Curve changes in the numerical simulation are almost non-existent and cause very little change in the energy absorbed [24,27]. The model used the MAT_24 (modified segmented linear plasticity) material model provided by LS-Dyna to describe the mechanical behavior of tubes and grids, which is applicable to most metallic materials. Since the rigid plate was not deformed during the test, it was modeled using the Mat_20 rigid material model. Previous studies have shown that aluminum alloys are insensitive to the material strain rate, so the effect of the material strain rate was not considered in the numerical analysis of this paper.

In order to better model the contact interactions between different components, three types of contacts were defined in the numerical model. The *CONTACT_AUTOMATIC_NODES_TO_SURFACE contact was used to define the contact between the thin-walled tube and the lattice structure to prevent the interpenetration of the structure during compression; the mutual contact interaction between the rigid wall and the hybrid structure was modeled by the *CONTACT_AUTOMATIC_NODES_TO_SURFACE contact algorithm; the *AUTOMATIC_SINGLE_SURFACE contact algorithm was used to define the bending contact between the thin-walled tube and the grid structure itself. The static and dynamic friction coefficients in all of the above contacts were set to 0.3 and 0.2 [24,25], respectively.

In order to avoid the influence of the mesh size on the accuracy of the numerical results, as well asthe excessive consumption of computational resources, the mesh convergence analysis was performed separately for the tube and lattice structures of the ULT1 hybrid structure, and the results are shown in Figure 8. For the shell cell of the thin-walled tube, five sizes (0.5 mm × 0.5 mm, 1.0 mm × 1.0 mm, 1.5 mm × 1.5 mm, 2.0 mm × 2.0 mm, 2.5 mm × 2.5 mm) were set for meshing; for the solid cell of the lattice structure, the meshes were set (0.25 mm × 0.25 mm, 0.3 mm × 0.3 mm, 0.35 mm × 0.35 mm, 0.4 mm × 0.4 mm, 0.45 mm × 0.45 mm). According to the figure, it can be seen that the 1.5 mm × 1.5 mm shell cell element and the 0.35 mm × 0.35 mm solid cell could ensure the accuracy of the numerical simulation and limit the increase in the computation time.

### 3.3. Impact Resistance Index

To investigate the crashworthiness of the hybrid structures under external loads, five crashworthiness metrics were used to evaluate the crashworthiness of the structures: energy absorption (EA), peak compressive force (PCF), mean compressive force (MCF), compressive load force efficiency (CFE), and specific energy absorption (SEA).

The PCF is defined as the peak force in the early compression phase of the force–displacement curve. For energy absorbers, the PCF should be minimized to achieve effective protection.

EA represents the total energy absorption consumed during compression and is defined as follows:(1)$$EA=\int_{0}^{d}F(x)dx$$where F(s) is the compression force in the direction of impact and d is the compression distance.

The mean compression force (MCF) is the average compression force over the compression distance and is calculated as follows:(2)$$MCF=\frac{EA}{d}=\frac{\int_{0}^{d}F(x)ds}{d}$$

To evaluate the loading efficiency and load carrying capacity, the compression load efficiency (CFE), derived from the ratio of the average compression load MCF to the peak compression load PCF, was considered, as in Equation (3). The closer the CFE is to 1, the more efficient the energy absorption and structural protection are.
(3)$$CFE=\frac{MCF}{PCF}\;\times \;100\%$$

Specific energy absorption (SEA) is defined as the ratio of absorbed energy (EA) to the mass of the structure and represents the energy absorption rate or energy absorption efficiency per unit mass, with m denoting the total mass of the structure.
(4)$$SEA=\frac{EA}{m}$$

For energy absorbers, the higher the SEA, the better the energy absorption. For structures with high SEA, reducing their mass can meet the design requirements related to light weight and energy efficiency when ensuring safety under crash conditions.

## 4. Results and Discussion

### 4.1. Quasi-Static Test Results

Figure 9 shows the quasi-static compressive axial deformation diagram of the experimental specimens, and the repeated experiments show the same deformation pattern. All specimens had similar deformation processes: as the extrusion proceeded, layer by layer collapses started due to buckling and local instability of the material. Progressive deformation patterns could be observed, and layer by layer stable collapse avoided the overall buckling of the structure.

Figure 9a gives the experimental deformation process of the hollow aluminum tube under continuous displacement, which gradually increased from the zero displacement of the specimen to the compaction stage, where the folds initially started from the position near the lower plate, and, after the first fold, the folds were superimposed on the previous one in turn and formed outwardly-folded flexural lobes that were alternately in the adjacent subunits, with less distal deformation and three mutually-folded folds formed throughout the compression process. Figure 9b and Figure 10c,d show the deformation patterns of the hybrid structures of ULT1, ULT2 and ULT3, respectively, where more folding junctions can be observed in the hybrid tubes compared to the single tubes due to the interaction between the thin-walled tubes and the internal lattice packing. The deformation of the hybrid structure of the ULT1 occurred first at the bottom, and as the compression proceeded, the two adjacent faces formed three and four. The folded regions and local failures on the prongs of the thin-walled tube formed cracks, which occurred because the lattice structure was so strong that structural tearing of the tube wall occurred when inward or outward folds were generated. The ULT2 and ULT3 structures differed from the initial deformation location of the ULT1 structure in that the structural deformation first occurred near the top. The ULT2 structure eventually formed four folds on each side. The ULT2 structure ended up with four folds on each side, while the ULT3 structure ended up with six mutually folded folds.

Figure 9e shows the quasi-static axial compression deformation process of the variable-angle-gradient lattice-filled structure VGLT. According to the literature [1], it is known that, in the variable-angle-gradient lattice structure, the specific strength of the cell element with a small angle between the supporting rods is lower than that of the large-angle cell element for the same supporting rod radius. Therefore, the deformation of the hybrid structure occurs first in the lower part of the lattice structure where the strength is lower, which is influenced by the local strength of the lattice structure. Figure 10f shows the deformation process of a thin-walled tube LGLT filled with a layered gradient lattice. Since the density of the lattice structure increases from top to bottom in order, the strength of the hybrid structure is also smaller at the top, and the deformation of the hybrid structure occurred first at the top.

When comparing the three hybrid structures with uniform density, the most obvious difference in the compression process is the number of folds formed in the deformation. With the increase in the number of lattice transverse cell elements, the number of folds gradually increased from three to six in the empty tube, and the lattice packing made the thin-walled tube deformation folds gradually unstable and irregular; when comparing the two gradient hybrid structures, they all gradually formed stable and continuous folds with the increase in the compression displacement. The final number of folds was the same as that of the ULT2 with the same number of transverse cell elements. Considering the inconsistent density distribution of the gradient lattice structure and the initial buckling, the interaction effect between the gradient lattice and the tube varied with the region, so that the thin-walled tube could produce a controlled and stable deformation pattern by changing the density distribution of the filled gradient lattice.

Figure 10 shows the force–displacement curves of the specimens obtained in the quasi-static experiments. All samples of all structures showed similar compression curves, which could be divided into three different characteristic phases. First is the initial linear elastic phase, where the compression force increases rapidly and reaches the peak load. After that, the load decreases significantly to reach the nonlinear response plateau phase, which represents the main energy absorption process of the energy absorber and is very important for energy absorbing structures in which the curve presents the formation of peaks and valleys and the formation of each fluctuation in the force–displacement curve corresponds to the formation of a fold. Finally, as the compression displacement increases gradually and reaches densification, the densification phase occurs, and the force increases rapidly with the increase in displacement.

The force–displacement curve of the uniform density lattice-filled tube was similar to that of the empty tube, where the compression force fluctuated almost around a constant value. Due to the enhancement effect of the lattice structure, the average compression force of the hybrid tube was much higher than that of the pure aluminum tube, and the number of folds formed increased with the increase in the number of cytons, thus resulting in an increase in the number of peaks and valleys and a smaller amplitude of the curve fluctuations. For the VGLT filled with a variable-angle gradient lattice, although the strength of the cell structure became larger due to the large angle, the relative density of the lattice in the same volume became smaller, the interaction force between the lattice and the thin wall decreased, and the amplitude of the compression force showed a gradually decreasing trend as the compression proceeded. However, for the LGLT tube filled with a graded lattice, due to the existence of cells with different radius rods in this graded structure, in the compression process, the deformation occurred layer by layer. First, in the upper less dense position, the performance of the graded structure and its load-bearing capacity were weakened by the layer of cells with fine rods. With the increase in the lattice density, the compression force had a gradually increasing trend, the formation of multiple peak regions occurred, and multiple peak regions were formed.

In general, all six types of thin-walled tubes were crushed in a similar way, i.e., symmetrically folded deformation, which formed regular and continuous folds. However, due to the influence of different forms of filling lattice structure, progressive folding deformation and fluctuation curves with different characteristics were formed.

### 4.2. Numerical Simulation and Experimental Comparison

In order to verify the accuracy of the numerical model, the quasi-static experimental results were compared with the numerical simulations. Figure 9 shows the quasi-static experimental and numerical simulations of the compression deformation process for the air tube and the five hybrid structures. It is clear that both the experimental and numerical simulations of the compression showed progressive folded pleat deformation and they had the same pleat flaps; additionally, the deformation patterns of the experiments and numerical simulations showed good agreement. In addition, the force–displacement curves of the experimental and FEM simulations also had similar trends, showing good agreement. At the start of the compression test, the indenter and specimen did not fit perfectly at the beginning, but, in the numerical model, the indenter and specimen fit perfectly. As a result, the measured initial stiffness was lower than the numerical calculation, which led to a delay in deformation for some of the experimental curves in the graph. The PCF values for the simulated curves were all higher than the experimental curves and had an error of less than 7%, due to the numerical simulation being set at a speed of 1000 mm/s, which is much greater than the actual compression test speed. Numerical integration of the experimental and simulated force and displacement curves to obtain values of energy absorption resulted in an error of less than 9% for all models. The difference between the experiments and simulations may have also been due to measurement errors and manufacturing accuracy, as can be seen in Table 3. In Section 2.2 where there was a difference between the mass of the printed specimen and the design value, and the smaller the supporting rod radius was, the greater the increase in specimen mass over the design value and the greater the error, so that the energy absorption values obtained from the experiments and finite element simulations were not always equal for a given compression distance. The experimental value of energy absorption for the ULT1 was 7.93% smaller than the simulated value, the experimental value for the ULT2 structure was approximately equal to the simulated value with an error of 2.91%, and the experimental value for the ULT3 hybrid structure was 4.83% larger than the simulated value. The above indicates that the numerical modelling approach is accurate enough to predict the deformation pattern and force response of the hybrid structure well. It should be noted that, at the end of the compression distance to reach densification, there was a difference in the effective compression distance between some of the experimental and numerical curves, so it was difficult for the numerical simulation to characterize the compression behavior of the hybrid tube during the densification stage.

Figure 11 shows the finite element structural sections of the empty tube and the ULT2 structure, which shows the number of folded flaps and half-wavelength of the structure. The number of folded flaps in the ULT2 increased from three to four in the empty tube, and the half-wavelength of the hybrid structure was significantly shorter than that of the empty tube at the same compressional displacement, which was measured as λ1 = 15.32 mm > λ2 = 10.56 mm. This is because the coupling between the lattice and coupling between the thin-walled tube inhibits the inward folding of the tube and the expansion of the folded wavelength in the tube wall, thus forming more plastic hinges and absorbing more energy during the deformation, while the densification phase of the hybrid structure occurs earlier. In fact, by observing the experimental and simulated deformation diagrams of the hybrid structure, it can also be found that the number of folded flaps increased with the increase in the number of cytons in the transverse section, and the wavelength decreased with the increase in the number of cytons.

### 4.3. Crashworthiness

In order to compare the effect on the performance of the hybrid structure under different lattice arrangement forms at the same density, the impact resistance index of the hybrid structure was investigated. The experimental and numerical simulations of the impact resistance index values for the hybrid structure are presented in Table 4. The experimental data were generated by the testing machine at the compression distance d = 75 mm and its corresponding reaction forces, and the experimental values of EA and SEA were averaged. Figure 12 shows the data provided in Table 4. It is clear that the PCF of several models was very close to that of the empty tube, all within the range of 7.55 kN ± 0.35 kN, which indicates that the PCF was insensitive to the lattice structure filling and its arrangement form. The other four impact resistance indexes (EA, SEA, MCF, CFE) were all larger than that of the empty tube, and the variations were more obvious, indicating that the lattice filling and its arrangement order significantly affected these four indexes. According to the calculation equation of impact resistance, it can be seen that EA, SEA, and MCF had the same trend of variation, because several hybrid tubes had the same mass and compression distance, among which the EA (429.78 J), SEA (7.87 J/g), and MCF (5.73 kN) of the ULT3 were the highest among all hybrid structures. The CFE represents the load uniformity of the compression process, and hybrid tubes have higher CFE values. Specifically, the CFE of several hybrid structures increased by 56.07%, 81.59%, 134.27%, 86.74% and 84.45%, respectively, compared to the aluminum tube. It is clear that the rate of increase of the MCF was greater than that of PCF, thus resulting in higher CFE metrics for the hybrid tubes.

For the hybrid structure ULT with a uniform density lattice, according to the comparison results of the ULT1, ULT2, and ULT3 data, it can be seen that, although the relative densities of the three structures were consistent and all uniformly distributed, the differences in the structural arrangement caused the pore variation of the structure, and the EA and SEA of models ULT1, ULT2, and ULT3 were higher than those of the aluminum tubes by 14.42%, 36.05% and 83.02%. For the gradient hybrid structure, the EA and SEA of the VGLT and LGLT were 41.86% and 35.81% higher than those of the aluminum tubes, respectively. When comparing the impact resistance indexes of the ULT2, LGLT, and VGLT, although the difference of the PCF was small, the PCF of the LGLT hybrid structure was the smallest, because the density distribution of the lattice structure was different, and the LGLT had the smallest density lattice structure. The SEA of the three hybrid structures showed the following trend: VGLT > ULT2 > LGLT. The results of two impact resistance indexes, SEA and MCF, for the ULT2 and LGLT structures were similar and 13% smaller than VGLT. The energy absorption characteristics of the hybrid structure with aluminum foam reinforced thin-walled tubes were compared and discussed, and it was found that the hybrid structure designed in this paper improved the SEA by 52.5% over the conventional structure.

Experimental and numerical simulations of quasi-static compression, as well as the analysis of crashworthiness metrics, showed that lattice filling can increase the load carrying capacity, energy absorption, and specific energy absorption of a pure aluminum tube while increasing the CFE value with a small increase in the initial peak compression force. When hybrid structures are applied in crash situations, it is important not to increase the initial force experienced by passengers while increasing the specific energy absorption of the structure, as this may cause more damage. When comparing the various crashworthiness metrics of the hybrid tube, it can be seen that the ULT3 was considered to be the best design for the lattice filled structures investigated in this study. This finding also provides a valid guide for designing hybrid structures with higher EA, SEA, CFE, and lower PCF.

### 4.4. Lattice–Shell Interaction Effect

In this subsection, the effect of lattice packing on the energy absorption characteristics of the hybrid structure in quasi-static compression mode is investigated. The energy absorption of the lattice-filled structure consists of three components:(1)The average compression breaking force of the tube without lattice packing.(2)The average crushing force of the lattice packing.(3)The interaction effect between the tube and the lattice.

However, the contribution of these components to the energy absorption capacity is not clear. In order to better understand the interaction effects induced by lattice packing, the impact resistance of individual lattices under axial compression was numerically calculated, and the load–displacement curves and energy absorption curves of empty tubes, lattices only, the sum of empty tubes and lattices, and several hybrid structures were plotted to investigate the effect of lattice packing on the impact resistance of hybrid structures.

It is clear from Figure 13 that the area under the force–displacement curves of all the hybrid structures was much larger than that under the sum of empty tube + lattice curves, and that the energy absorbed by the hybrid tubes was greater than the sum of a single thin-walled tube and a single lattice. It is well known that the thin-walled tube had higher stiffness and strength. A comparative analysis shows that the energy absorbed by the aluminum tube was much larger than that absorbed by the lattice, and when the deformation was small, the energy was almost completely absorbed by the square tube. When compared with the square hollow tube at the early stage of compression, the lattice structure could not provide sufficient load support due to the absence of any vertical member. The load displacement curve of the BCC lattice structure had no obvious peak and showed smooth and continuous compression behavior, and the lattice structure had a long plateau area, which is very similar to the compression behavior of the foam structure.

In Figure 13, the pink shaded region indicates the energy absorption due to interaction effects, and the shaded region increased with compressional displacement. The square tube imposed additional lateral constraints on the lattice structure compared to the lattice under uniaxial compression. The role of the lattice in the hybrid structure is to inhibit the flexural deformation of the aluminum tube and to increase the energy absorption by increasing the number of plastic hinge folds in the thin-walled tube. As a positive feedback, the square tube in the hybrid structure dissipates more impact energy than expected than the square tube alone. Both the energy absorbed by the lattice structure during compression and the interaction effect are smaller than those absorbed by the square tube, which remains the main component of energy absorption.

Figure 14 shows the percentage contribution of each component, and the energy absorption is ranked by the percentage contribution as: tube > lattice structure > interaction effect. The ULT3 has the highest contribution of interaction effects among all hybrid structures with 43.4% for ULT3, 26.68% and 28.97% for ULT2 and ULT1 structures, and 27.42% and 31.83% for VGLT and LGLT, respectively. Furthermore, when comparing the interaction contribution between different types of hybrid tubes, the difference in the interaction between structures with different density distributions for thin-walled tubes was caused by the interaction effect coming from both the number of lattice supporting rods in contact with the tube wall and the variation of the lattice deformation model. By comparing the uniform density hybrid structure, the ULT2 with the gradient density hybrid structures VGLT and LGLT, it can be seen that the gradient structure had a higher interaction effect than the uniform density structure, which indicated that the gradient-filled hybrid tube had a higher energy absorption potential. Therefore, there is still room to improve the energy absorption capacity of the overall structure by improving the interaction between the components, and to further explore the enhancement effect of the lattice on the aluminum tubes, parametric studies were carried out in the subsequent sections.

## 5. Parametric Optimization Analysis

In this section, in order to gain a comprehensive understanding of the axial compression performance of the lattice-filled thin-walled tube hybrid structure, a parametric analysis was performed using numerical simulations to investigate the effects of tube wall thickness, lattice density, and gradient density distribution on the crashworthiness of the hybrid structure. The parametric analyses were performed using the numerical model in Section 3.3 with quasi-static loading conditions. In these analyses, the dimensions of the thin-walled aluminum tubes were kept constant: length L = 120 mm and width b = 30 mm. The other parameters were kept constant when the effect of one parameter was studied. It should be noted that, in the course of the parametric study, initial defects were introduced in the hybrid tube as triggers in order for the structure to develop a stable deformation.

### 5.1. Effect of Tube Wall Thickness

The thickness of thin-walled tubes is a key factor affecting the impact resistance of hybrid structures. In this section, four different wall thicknesses, t = 0.5 mm, 1 mm, 1.5 mm, and 2 mm, were selected to investigate the effect of wall thickness t on the impact resistance performance of several hybrid structures. After numerical calculation, the impact resistance index was calculated at δ = 75 mm. Table 5 shows the detailed data of the crashworthiness indexes, and the trends of SEA, MCF, PCF, and CLE with wall thickness are shown in Figure 15. It can be seen that, as the tube wall thickness increased, the peak initial impact force increased significantly, while the energy absorption, specific energy absorption and load-bearing capacity of the structure were greatly enhanced, all due to the more severe plastic deformation caused by the larger mass of the structure. For example, the ULT2 hybrid structure with a tube thickness of 2 mm was 632% of the energy absorbed, 232% of the specific energy absorption, and 633% of the load carrying capacity of a hybrid structure with a tube thickness of 0.5 mm. In addition, the CLE also showed better results at larger thicknesses and increases with the number of transverse cells.

When comparing the crashworthiness indexes of ULT hybrid tubes filled with uniform density lattices at different wall thicknesses, it can be found that the changes in SEA and PCF showed the same approximately linear growth pattern as the thickness increases, and the crashworthiness indexes of all three hybrid tubes were ULT1 > ULT2 > ULT3. In the design of energy absorbers, high PCF should be avoided; therefore, in order to improve the crashworthiness of ULT hybrid tubes, it is possible to increase the structure thickness within the PCF range. Therefore, in order to improve the impact resistance of the ULT hybrid tube, the wall thickness of the structure can be increased appropriately within the range of the PCF. When comparing the impact resistance indexes of ULT2, VGLT and LGLT for hybrid tubes with gradient density lattices and uniform density lattices at different wall thicknesses, the trend of the PCF of the three structures was relatively stable and increased approximately linearly with the increase in wall thickness, and the PCF of hybrid tubes with the same wall thickness was always LGLT > ULT2 > VGLT. However, the trends of SEA and CLE for both gradient structures showed a high degree of nonlinearity, with the SEA of LGLT gradually changing from maximum to minimum as the wall thickness increased, with the CFE of ULT2 being the maximum and the LGLT the minimum at lower wall thickness, while the CLE of the VGLT hybrid tube gradually increased to maximum as the wall thickness increased. The above study shows that the requirements of maximum SEA and minimum PCF for hybrid tubes have extensive conflicts in terms of tube thickness and can be further optimized to obtain maximum SEA and minimum PCF and keep the CFE at a high level accordingly.

### 5.2. Effect of Lattice Density

In order to investigate the effect of lattice density on the energy absorption performance of the hybrid structures, the ULT2 was selected as the object of study, and four hybrid structures with lattice diameters of 1 mm, 1.5 mm, 2 mm, and 2.5 mm, named ULT2-a, ULT2-b, ULT2-c, and ULT2-d, respectively, were selected, where the thickness of the aluminum tube was set to 1 mm. Since the models ULT2-a and ULT2-d had the minimum and maximum lattice density, respectively, this may have affected the compression efficiency: (δ is the maximum compression length at the densification point and L is the length of the tube). Theoretically, in order to determine Se, the maximum compression length should be determined, and an efficient way to determine its is to use the deformation efficiency–displacement diagram presented in [1], where the energy absorption efficiency is defined as:(5)$$\eta =\frac{\int_{0}^{\delta }F(s)ds}{F_{\max}}$$
where *s* and *F* (*s*) are the compression distance and force, respectively, and *F*_max_ represents the maximum compression load other than the peak compression force in the interval [0, d]. For a tubular structure, the compression increases as compression proceeds until it reaches its peak value, and then decreases rapidly as the structure densifies, and the corresponding compression displacement can be considered as the densification displacement. Figure 16 shows the force-displacement and deformation efficiency–displacement plots. According to Figure 17 and the definition of deformation efficiency in the equation, the maximum compression lengths δ for the four hybrid structures were determined as 84.8 mm, 79.2 mm, 68.7 mm, and 62.4 mm, respectively. Therefore, the compression efficiency Se for models ULT2-a, ULT2-b, ULT2-c, and ULT2-d were calculated as 70.7%, 66.1%, 57.2%, and 52.1%, and the compression efficiency of ULT2-d was the smallest among these models.

Table 6 shows the crashworthiness data for the hybrid tubes with different lattice densities. It is obvious from the figure that the higher the density of the lattice, the higher the initial peak force PCF. However, for the EA, SEA, MCF, and CFE, four impact resistance indicators were further compared: ULT-a < ULT-b < ULT-c > ULT-d. That is, it increased with increasing lattice density within a certain range, because when the density of the lattice was higher, the structural yield force was also higher, which led to a stronger inhibition of buckling of the aluminum tube. In other words, for higher density lattice fillings, more energy is required for flexural deformation of the aluminum tube, the interaction effect is significantly enhanced, and the energy absorption capacity of the whole structure is improved. However, too high density will lead to smaller effective compression distance and lower impact resistance indexes, for example, the impact resistance indexes of ULT2-d were not as excellent as those of ULT2-c. Therefore, the increase in PCF and the decrease in impact resistance performance brought by too high lattice density should be avoided. From the above analysis, it can be concluded that changing the filling density is an effective way to design excellent energy absorbers with high energy absorption efficiency.

### 5.3. Variable Angle Gradient Stratification Strategy

From Section 5.1, it can be seen that the variable angle gradient lattice had better specific energy absorption and lower peak stress when the wall thickness was smaller. In this subsection, the crashworthiness of the VGLT structure under different layering strategies was investigated by combining different angular cell structures to form a gradient lattice with different densities and topologies while maintaining the same supporting rod diameter. As is shown in Figure 17, the periodic lattice structure with different layering strategies, the number of transverse cytosol elements *m* = 2, and the number of longitudinal layers *n* = 8, 7, 6 and 5 were named VGLT-5, VGLT-6, VGLT-7 and VGLT-8, respectively, where the height of each layer was in an equal series with the previous layer, as shown in Table 7 for each layer, and the thickness of the thin-walled tube wall was specified as t = 1 mm.

Table 8 shows the detailed data of the impact resistance analysis: as the number of layers decreased, the mass of the hybrid structure decreased, but the initial peak impact force increased, and the EA, SEA, MCF, and CFE decreased simultaneously. As the lattice angle became larger, the number of layers n of the lattice decreases, resulting in fewer contact points between the lattice and the thin wall, which reduces the mutual contact area and the interaction effect. From Section 4.3, it can be seen that the interaction contributed more to the energy absorption of the hybrid structure, and reducing the contact points made the contribution of the interaction to the energy absorption much lower. This indicates that increasing the contact points between the lattice and the tube wall can enhance the energy absorption performance of the hybrid structure, and the law of the number of contact points on the energy absorption effect of the hybrid structure needs to be further studied.

### 5.4. Variable Density Gradient Stratification Strategy

To further understand the effect of gradient distribution on improving the crashworthiness of LGLT hybrid structures, several gradient lattices with different density distributions were designed to investigate the response of hybrid structures to impact loads by numerical methods. The lattice structure has increasing density from top to bottom, where the largest supporting rod diameter D and the smallest diameter d is called the gradient coefficient, which is the ratio of d to D: ζ = d/D. In this section, four LGLT hybrid structures with different gradient density distributions, named LGLT1, LGLT2, LGLT3, and LGLT4, were studied, and the model schematic diagrams of the four gradient lattice structures are shown in Figure 18. The bottom lattice has a constant diameter D = 4 mm and ζ = 1, 0.75, 0.5, 0.25. The tube thickness is chosen to be t = 1 mm and the tube width was 30 mm.

The force–displacement curves as well as the energy absorption–displacement curves of the four structures under axial compression loading are shown in Figure 19. Table 9 shows the data for the impact resistance index. it is clear that the gradient played a dominant role in the force–displacement and in the EA–displacement curves of the hybrid structure. As ζ decreased, the plateau forces in the force–displacement curves increased, thus leading to a substantial increase in the energy absorption of the hybrid structure. The smaller ζ was, the later the densification appeared, which was due to the increase in the total mass of the lattice structure as ζ increased, thusleading to a decrease in the maximum compression distance ξmax. In contrast, the smaller ζ the smaller the PCF, because there is a gradient change in density, which is smaller in the upper part. Considering the effect of structural mass, the SEA was compared and it was found that structures with a large gradient within a certain range performed better. In general, considering the above indicators, a smaller gradient factor tends to reduce the PCF, which is preferred when designing energy absorbers, and a smaller gradient factor within a certain range can improve the impact resistance of thin-walled structures.

## 6. Conclusions

In this study, lattice structures with different pore and gradient densities were created to fill thin-walled aluminum tubes by changing the geometric parameters of the BCC lattice cells (including the radius of the supporting rod, pinch angle, and number of transverse cells), and two types of lattice structures with uniform and gradient densities were designed to enhance the energy absorption performance of the thin-walled aluminum tubes to achieve diverse mechanical responses and a wide range of mechanical properties of the hybrid structure. The deformation mode and mechanical response of the hybrid structure were analyzed through experimental tests and numerical simulations, and the impact resistance behavior of the hybrid tube under axial load was investigated in depth. Several main conclusions were obtained as follows:

(1) The EA and MCF of the hybrid structure were found to be much higher than those of the empty tube by quasi-static compression experiments and numerical simulations, the folding wavelength of the hybrid structure was significantly shorter, and the number of folding was higher, while the initial peak force PCF was not significantly higher than that of the empty tube, while SEA and CFE could be enhanced by up to 83.02% and 134.27%, respectively, compared with the empty tube. There are differences in the interaction of different lattice distributions on thin-walled tubes, and the gradient density hybrid structure has a higher interaction effect than the uniform density hybrid structure. The energy absorption characteristics of the hybrid structure with aluminum foam reinforced thin-walled tubes were compared and discussed, and it was found that the hybrid structure designed in this paper improved the SEA by 52.5% over the conventional structure.

(2) For uniform density hybrid tubes, the number of transverse cells affects the formation of the number of folds, and the number of folds gradually increased from four to six as the number of transverse cells increased from one to three. Meanwhile, the impact resistance indexes also increased simultaneously with the increase in the number of transverse cells, with the highest increase of 59.95% in SEA.

(3) For the VGLT filled with a variable-angle gradient lattice, the amplitude of compression force showed a gradual decrease. However, for LGLT tubes filled with a layered gradient lattice, the compression force showed a gradual increasing trend and formed multiple peak regions. the PCF of the LGLT hybrid structure was the smallest, and the SEA and MCF results of ULT2 and LGLT structures were similar and 13% smaller than those of VGLT.

## Figures and Tables

**Figure 1 materials-16-01871-f001:**
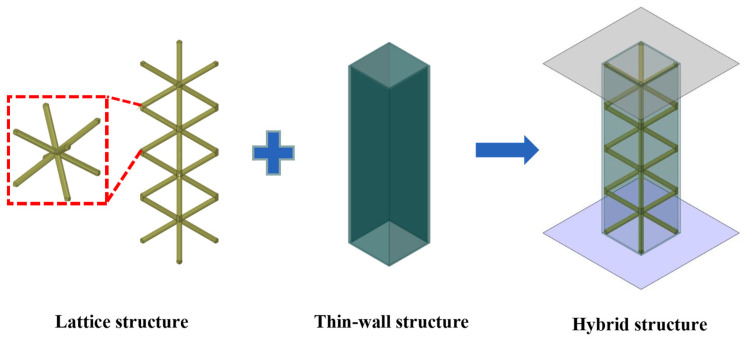
Schematic diagram of hybrid structure.

**Figure 2 materials-16-01871-f002:**
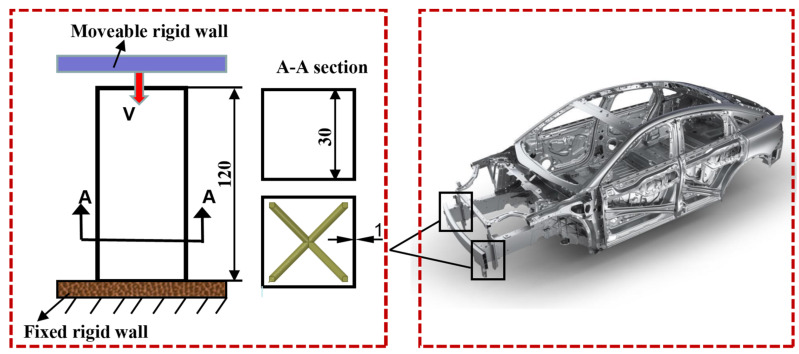
Schematic diagram of a simplified model of a car energy absorption box.

**Figure 3 materials-16-01871-f003:**
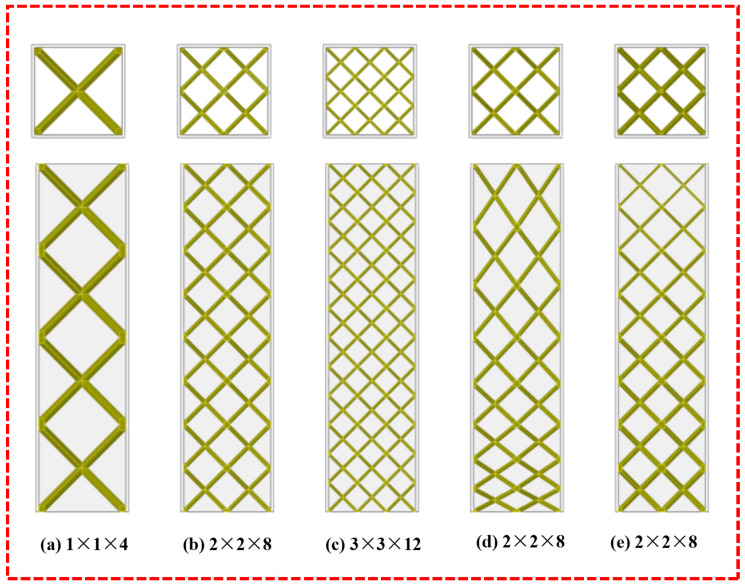
Main and top views of the lattice structure with different cell arrangements.

**Figure 4 materials-16-01871-f004:**
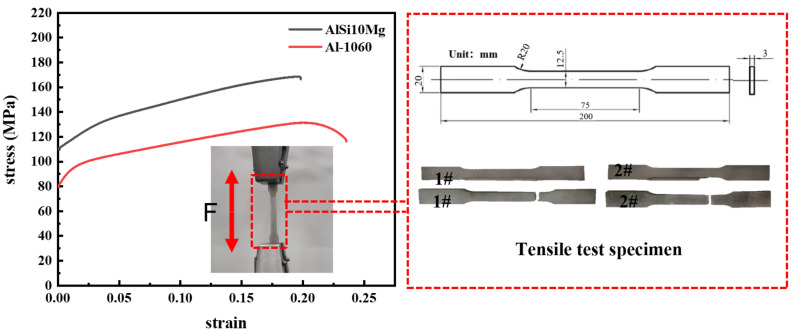
True stress–strain curves of the materials.

**Figure 5 materials-16-01871-f005:**
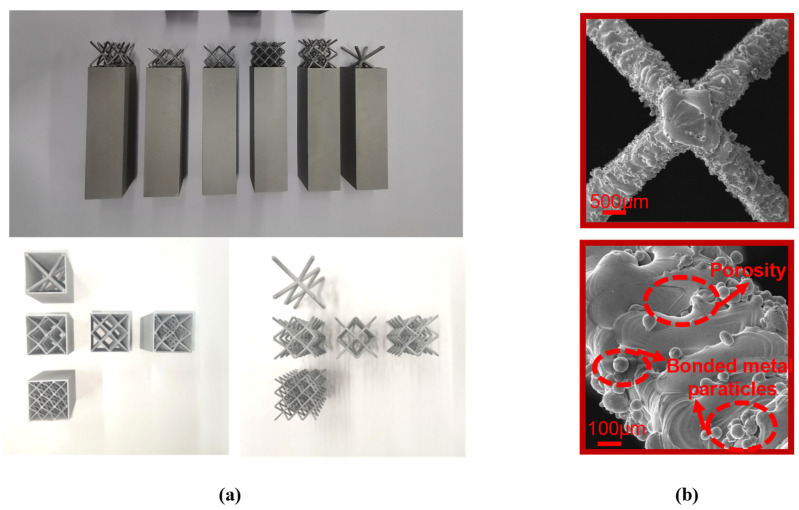
(**a**) Macroscopic morphology of the experimental specimen; (**b**) high magnification SEM image of the surface morphology of the sample.

**Figure 6 materials-16-01871-f006:**
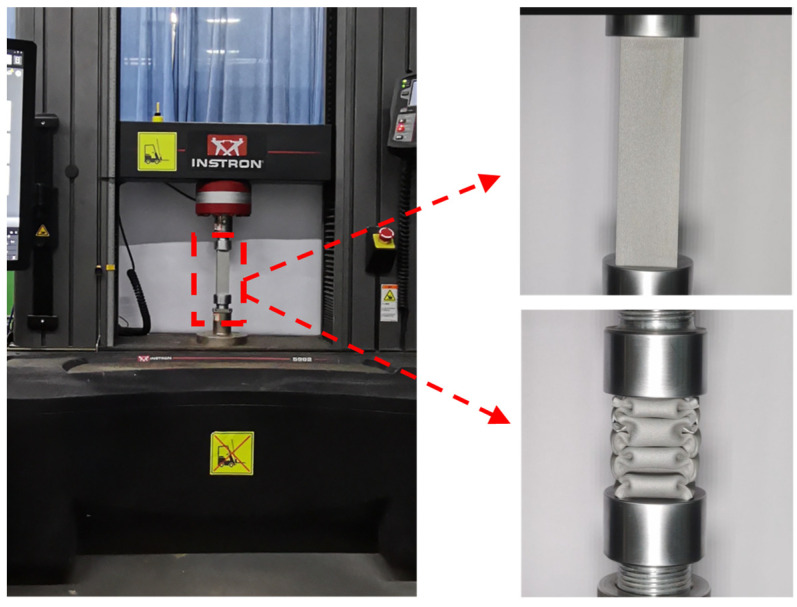
Quasi-static compression experimental system.

**Figure 7 materials-16-01871-f007:**
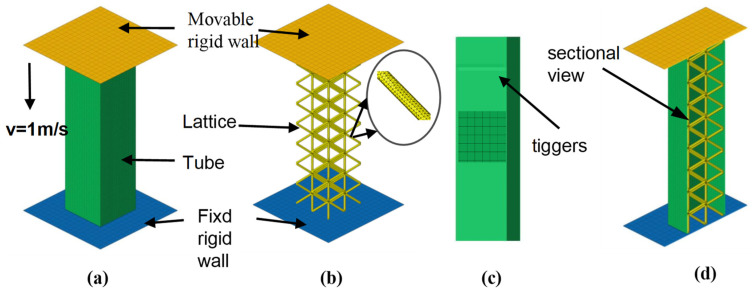
Finite element model of the hybrid structure (**a**) Thin-walled tube finite element model (**b**) Lattice finite element model (**c**) Tiggers (**d**) Finite element model section.

**Figure 8 materials-16-01871-f008:**
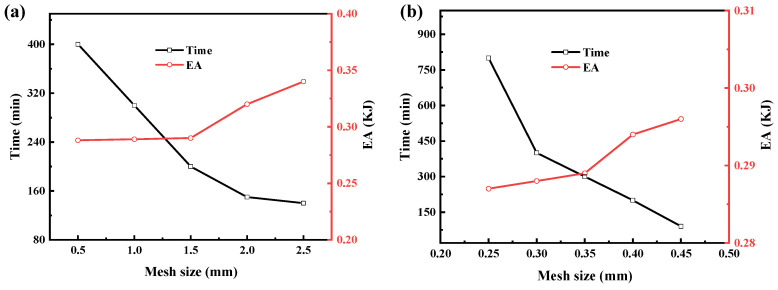
Convergence analysis of the finite element model: (**a**) convergence analysis of thin-walled aluminum tubes, (**b**) convergence analysis of dotted structure.

**Figure 9 materials-16-01871-f009:**
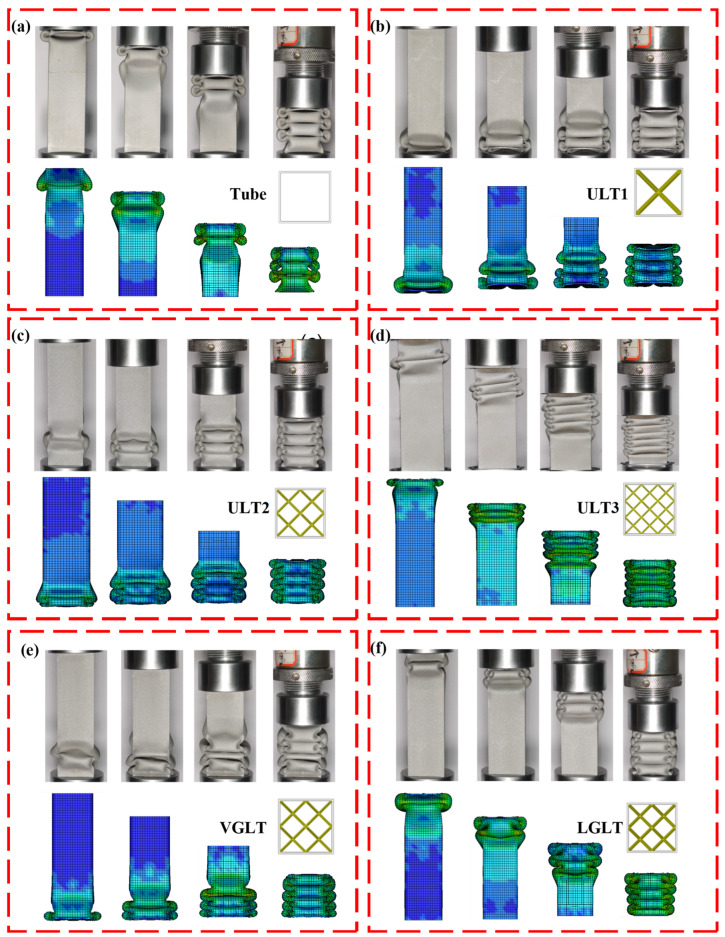
Deformation process of specimen experiment and simulation: (**a**) ULT1, (**b**) ULT1, (**c**) ULT2, (**d**) ULT3, (**e**) VGLT, (**f**) LGLT.

**Figure 10 materials-16-01871-f010:**
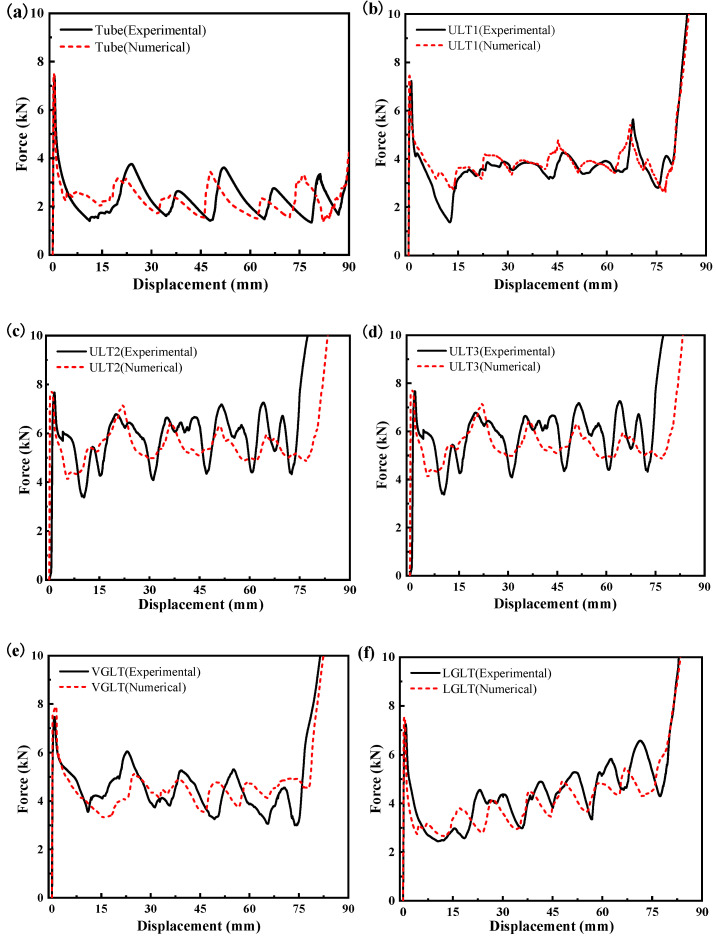
Force–displacement curves of specimen experiments and simulations: (**a**) ULT1, (**b**) ULT1, (**c**) ULT2, (**d**) ULT3, (**e**) VGLT, (**f**) LGLT.

**Figure 11 materials-16-01871-f011:**
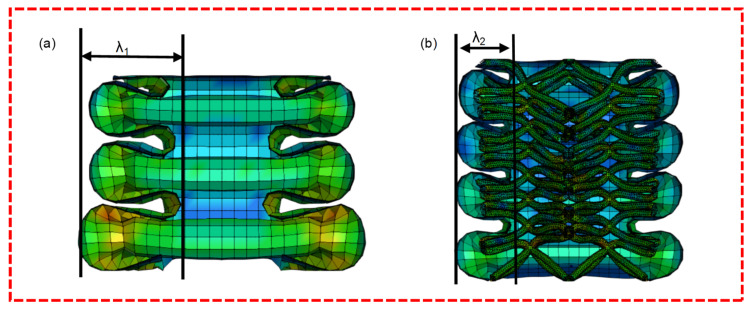
Numerical simulation cross section of air tube and ULT2: (**a**) Air tube structure profile, (**b**) ULT2 hybrid structure profile.

**Figure 12 materials-16-01871-f012:**
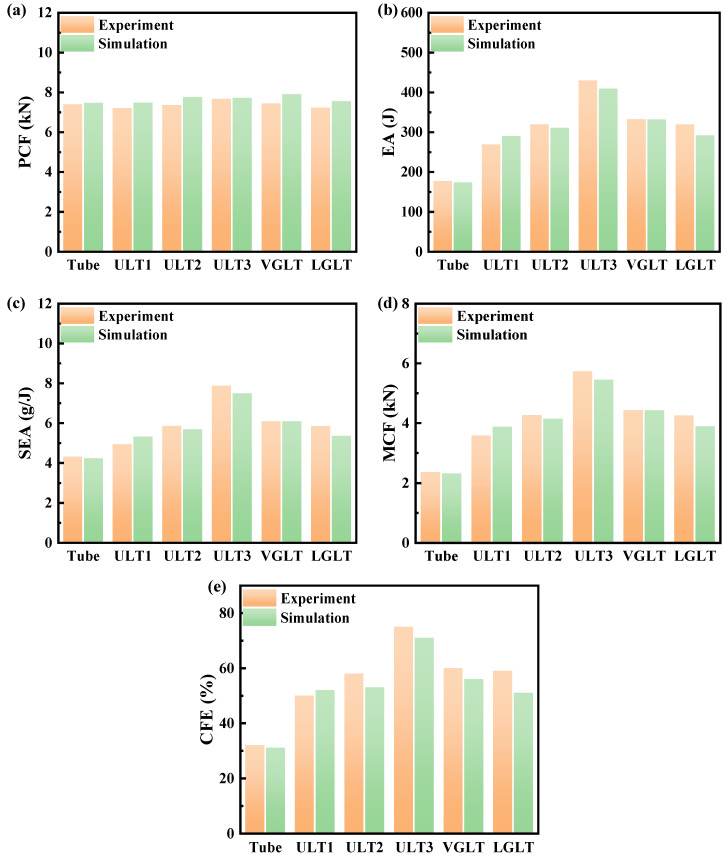
Bar charts of crashworthiness indexes for empty and hybrid tubes (**a**) PCF; (**b**) EA; (**c**) SEA; (**d**) MCF; and (**e**) CFE.

**Figure 13 materials-16-01871-f013:**
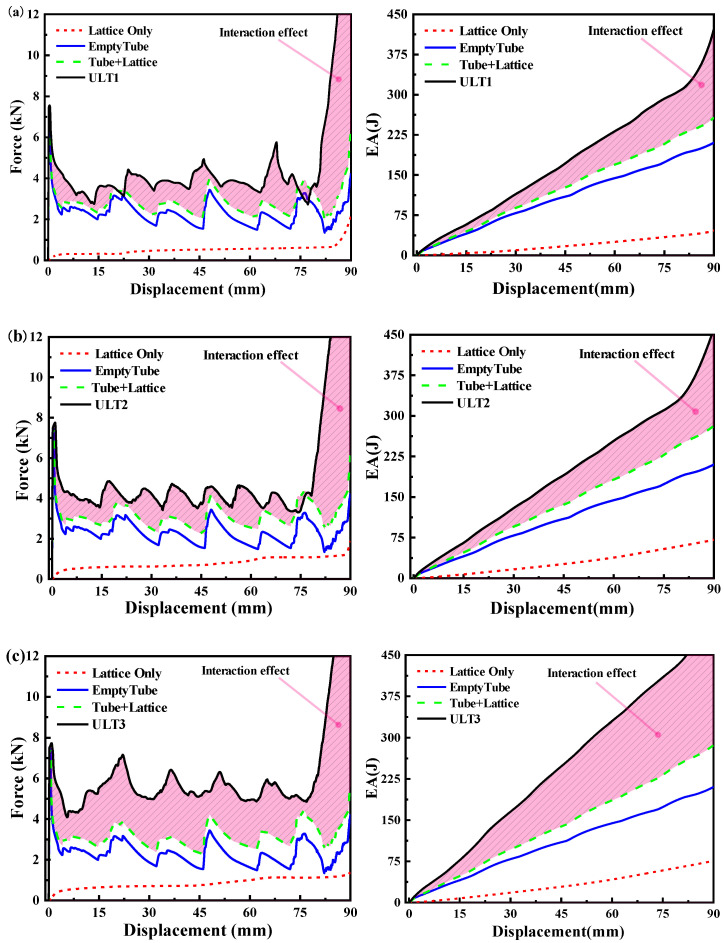
Force–displacement curves and energy absorption of each component of the hybrid structure: (**a**) ULT1, (**b**) ULT2, (**c**) ULT3, (**d**) VGLT, (**e**) LGLT.

**Figure 14 materials-16-01871-f014:**
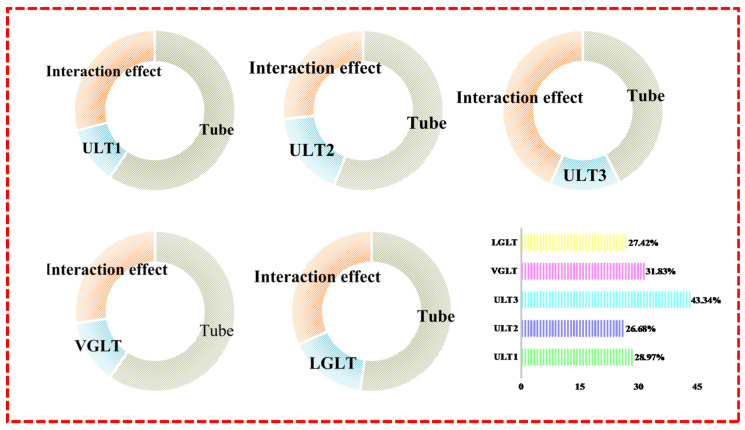
Contribution of energy uptake.

**Figure 15 materials-16-01871-f015:**
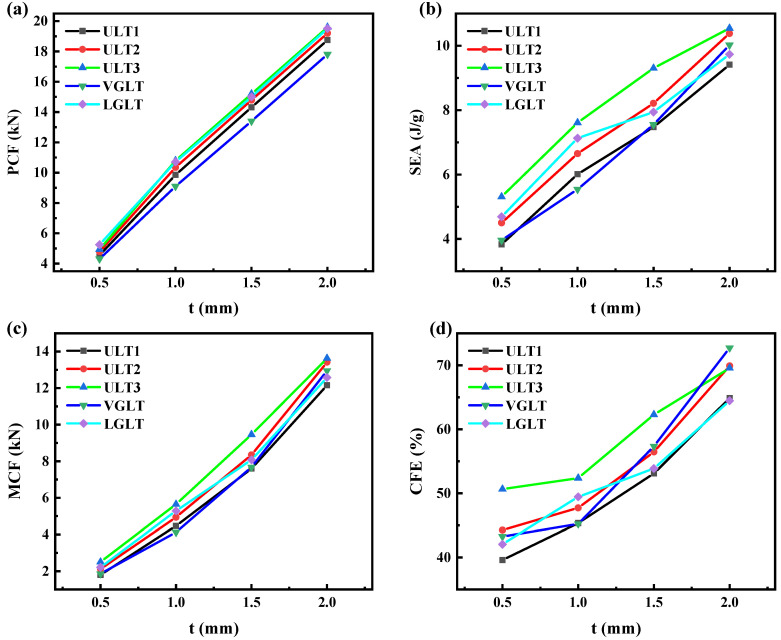
Effect of wall thickness on impact resistance of hybrid structures (**a**) PCF; (**b**) SEA; (**c**) MCF; and (**d**) CFE.

**Figure 16 materials-16-01871-f016:**
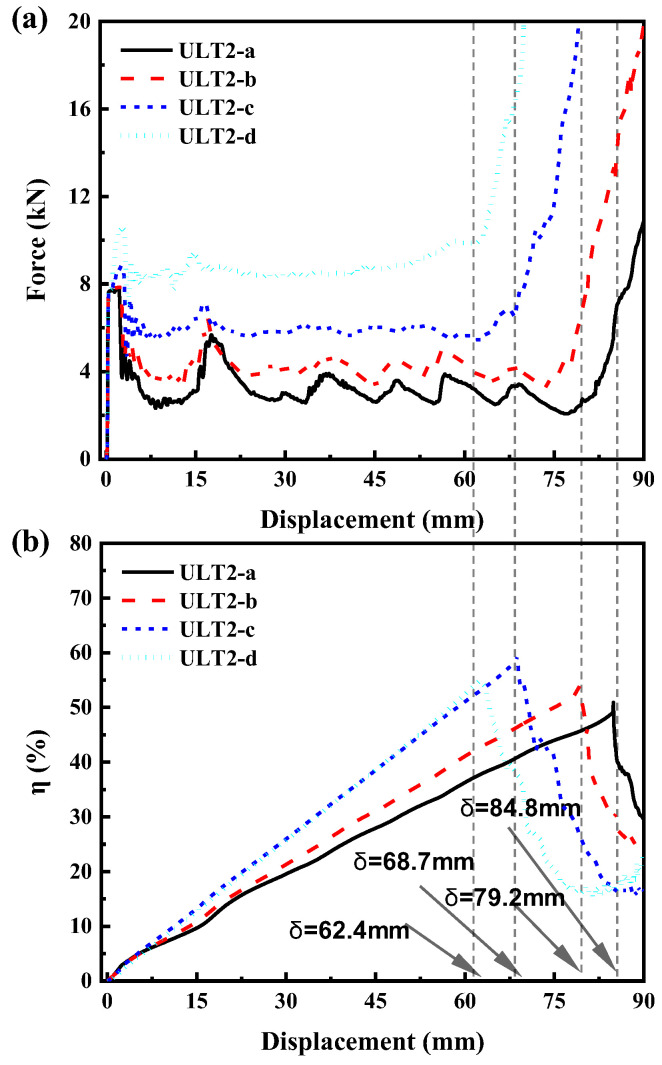
(**a**) Force–displacement curves for the hybrid structural model and (**b**) deformation efficiency-displacement curves for the hybrid structural model.

**Figure 17 materials-16-01871-f017:**
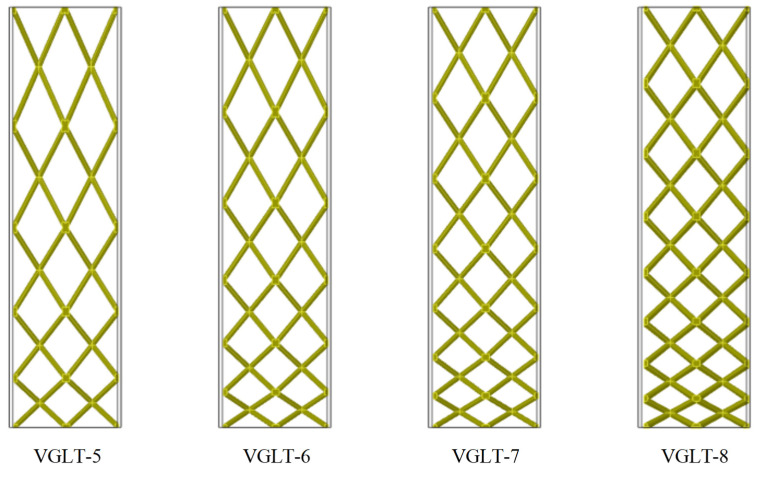
Lattice structure model inside the VGLT hybrid structure with different gradient configurations.

**Figure 18 materials-16-01871-f018:**
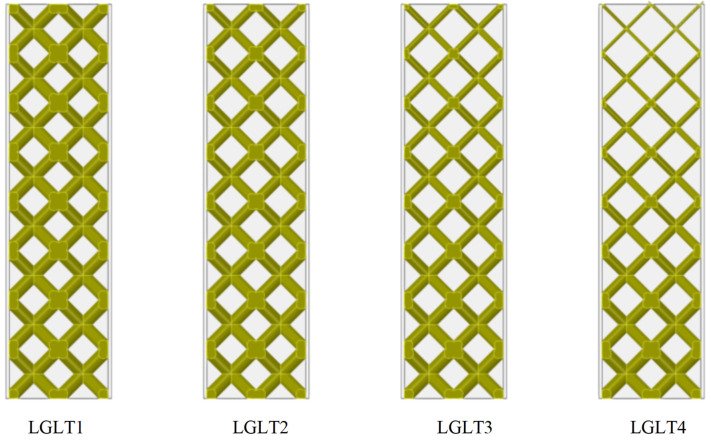
Internal lattice structure model of LGLT hybrid structure with different gradient configurations.

**Figure 19 materials-16-01871-f019:**
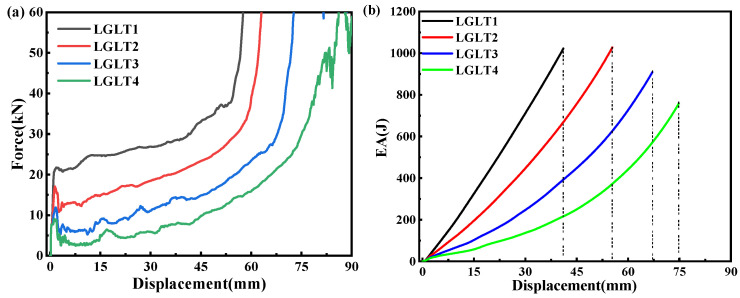
(**a**) Force–displacement curves (**b**) Energy absorption–displacement curves of four structures under axial compression load.

**Table 1 materials-16-01871-t001:** Material composition of AlSi10Mg (wt. %).

AlSi10Mg									
Al	Si	Fe	Cu	Mn	Mg	Ni	Zn	Pb	Ti
Balance.	9–11	0.055	0.05	0.45	0.2–0.45	0.05	0.1	0.05	0.15

**Table 2 materials-16-01871-t002:** Material properties.

	Density (g/cm^3^)	Elastic Modulus (GPa)	Poisson’s Ratio	Initial Yielding (MPa)	Final Yielding (MPa)	Ultimate Strain
Al-1060	2.69	69.60	0.33	80.68	130.25	0.20
AlSi10Mg	2.68	63.70	0.32	110.35	170.62	0.25

**Table 3 materials-16-01871-t003:** Weight of SLM printed specimens.

Types	ULT1	ULT2	ULT3	VGLT	LGLT
Design weight (g)	14.51	14.51	14.51	14.51	14.51
Measuring weight (g)	15.51	15.96	16.58	15.88	16.06

**Table 4 materials-16-01871-t004:** Number of impact resistance indexes of experimental specimens.

Samples	Cases	Δ_e_ (mm)	MCF (kN)	PCF (kN)	EA (J)	SEA (J/g)	CFE (%)
Tube	Exp.	75	2.36	7.39	176.73	4.30	31.89
	Num.		2.31	7.47	173.50	4.22	30.97
	Error (%)		−2.12	1.08	−1.83	−1.86	−2.88
ULT1	Exp.	75	3.58	7.21	268.76	4.92	49.77
	Num.		3.87	7.48	289.99	5.31	51.69
	Error (%)		8.10	3.74	7.90	7.93	3.86
ULT2	Exp.	75	4.26	7.35	319.23	5.85	57.91
	Num.		4.14	7.76	310.16	5.68	53.29
	Error (%)		−2.82	5.58	−2.84	−2.91	−7.98
ULT3	Exp.	75	5.73	7.67	429.78	7.87	74.71
	Num.		5.45	7.72	409.12	7.49	70.66
	Error (%)		−4.89	0.65	−4.81	−4.83	−5.42
VGLT	Exp.	75	4.43	7.44	332.28	6.08	59.55
	Num.		4.42	7.90	331.80	6.10	56.00
	Error (%)		−0.23	6.18	−0.14	0.01	−5.96
LGLT	Exp.	75	4.25	7.23	318.94	5.84	58.82
	Num.		3.89	7.55	291.43	5.34	51.47
	Error (%)		−8.47	4.43	−8.63	−8.56	−9.50

**Table 5 materials-16-01871-t005:** Energy absorption parameters with different wall thickness models.

Samples	Mass (g)	MCF (kN)	PCF (kN)	EA (J)	SEA (J/g)	CFE (%)	Thickness (mm)
ULT1	35.31	1.80	4.55	135.12	3.83	39.60	0.5
ULT2		2.12	4.79	159.00	4.50	44.26	
ULT3		2.50	4.94	187.61	5.31	50.64	
LGLT		2.21	5.25	165.56	4.69	42.05	
VGLT		1.86	4.31	139.83	3.96	43.26	
ULT1	55.71	4.47	9.85	335.02	6.01	45.35	1
ULT2		4.94	10.35	370.41	6.65	47.72	
ULT3		5.65	10.79	423.86	7.61	52.38	
LGLT		5.29	10.71	397.11	7.13	49.44	
VGLT		4.12	9.10	308.64	5.54	45.22	
ULT1	76.33	7.60	14.31	569.85	7.47	53.10	1.5
ULT2		8.35	14.79	626.34	8.21	56.47	
ULT3		9.46	15.19	709.61	9.32	62.29	
LGLT		8.08	15.00	605.96	7.94	53.86	
VGLT		7.68	13.40	576.31	7.55	57.34	
ULT1	96.94	12.16	18.75	912.268	9.41	64.87	2
ULT2		13.42	19.21	1006.56	10.38	69.90	
ULT3		13.63	19.59	1022.03	10.54	69.56	
LGLT		12.58	19.51	943.27	9.73	64.46	
VGLT		12.95	17.81	971.3	10.02	72.72	

**Table 6 materials-16-01871-t006:** Energy absorption parameters of the ULT2 model with different density configurations.

Samples	Mass (g)	Effective Compression Distance (mm)	PCF (kN)	MCF (kN)	EA (J)	SEA (J/g)	CFE (%)
ULT2-a	47.74	84.8	9.45	3.28	278.49	3.54	34.75
ULT2-b	55.71	79.2	10.35	4.29	339.95	5.16	41.47
ULT2-c	65.92	68.7	11.92	7.95	545.88	9.80	66.66
ULT2-d	78.61	62.5	13.26	6.61	412.51	8.64	49.85

**Table 7 materials-16-01871-t007:** Detailed data on the height of each floor of the VGLT hybrid structure.

Samples	Mass (g)	1 (mm)	2 (mm)	3 (mm)	4 (mm)	5 (mm)	6 (mm)	7 (mm)	8 (mm)
VGLT-8	54.61	8	10	12	14	16	18	20	22
VGLT-7	54.52	9	11	14	17	20	23	26	
VGLT-6	53.48	10	14	18	22	26	30		
VGLT-5	52.43	14	19	24	29	34			

**Table 8 materials-16-01871-t008:** Energy absorption parameters of the VGLT model with different gradient configurations.

Samples	Mass (g)	PCF (kN)	MCF (kN)	EA (J)	SEA (J/g)	CFE (%)
VGLT-8	54.61	10.71	5.29	397.00	7.27	0.49
VGLT-7	54.52	10.93	5.12	384.00	7.04	0.47
VGLT-6	53.48	11.01	5.00	375.00	7.01	0.45
VGLT-5	52.43	11.10	4.84	363.00	6.92	0.44

**Table 9 materials-16-01871-t009:** Energy absorption parameters of LGLT models with different gradient configurations.

Samples	Mass (g)	PCF (kN)	MCF (kN)	EA (J)	SEA (J/g)	Compression Distance (mm)
LGLT1	127.28	21.73	24.78	1021.52	8.03	41.1
LGLT2	109.89	17.04	19.88	1027.66	9.35	55.4
LGLT3	94.65	11.90	13.51	909.28	9.61	67.1
LGLT4	82.36	9.05	10.17	758.58	9.21	74.9

## Data Availability

Not applicable.

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
