# Peer review of "Crashworthiness Study of 3D Printed Lattice Reinforced Thin-Walled Tube Hybrid Structures"

_materials, 2023, doi:10.3390/ma16051871_

Round 1

Reviewer 1 Report

The presented paper deals with experimental and numerical investigation of compressive deformation characteristics of hybrid structures based on a hollow tube with 3D printed lattice filling.

Since optimization in the field of crashworthiness is very relevant, I consider the topic of the paper being highly interesting for the scientific community.

The paper is very well organized and written including detailed results section which is supporting the conclusions in a profound manner.

With respect to the overall level of the manuscript, I recommend its publication when the following minor comments are addressed:

1] Since the numerical simulations are fundamental part of the manuscript, I would like you to add more details regarding the FE mesh including visualization of the meshed bodies, number of elements/nodes in the simulations and the approximate calculation time.

2] Sensitivity of numerical results to mesh density has been performed and analyzed. However, in my experience with simulations of 3D printed lattices subjected to high overall compressive strain in LS-DYNA, the reliability of results may be substantially influenced also by the *SECTION_SOLID type, particularly when comparing the results of ELFORM4 vs. ELFORM17. You should comment on this.

3] In the description of the contact formulation, I would expect also a self-contact condition of both the lattice and the thin-walled body. However, these are omitted in the manuscript whatsoever. Could you provide further explanation?

4] The coefficients of friction are provided, but without reference to their source.

5] The quasi-static experiments were performed using the load rate of 2 mm/s, while the boundary condition in the numerical simulation is set to 1 m/s, i.e., 500 times higher load rate. The motivation for the selection of the velocity in the numerical simulations should be thoroughly explained, particularly with respect to the lack of experimental verification and the resulting strain rate effects in the lattice structure.

5] Review the captions of individual figures, where subfigures are present but without context in the caption itself (e.g., Fig. 8).

6] The deformation absorption characteristics of the structures are analyzed with regard to various aspects. However, I am missing cross-referencing of the obtained results with current state-of-the-art in the field, i.e., include a discussion section in your manuscript.

7] Schematics of the lattice and the thin-walled tube in Figure 1 might be confusing for the readers without cross-referencing throughout the manuscript. I recommend to rotate the visualization of the components around the vertical axis.

Author Response

Point 1: Since the numerical simulations are fundamental part of the manuscript, I would like you to add more details regarding the FE mesh including visualization of the meshed bodies, number of elements/nodes in the simulations and the approximate calculation time.

Response 1: Changes were made to Figure 7 to add a local enlargement of the mesh and to add the number of cells and the approximate time used for the calculation in the manuscript.

Point 2:  Sensitivity of numerical results to mesh density has been performed and analyzed. However, in my experience with simulations of 3D printed lattices subjected to high overall compressive strain in LS-DYNA, the reliability of results may be substantially influenced also by the *SECTION_SOLID type, particularly when comparing the results of ELFORM4 vs. ELFORM17. You should comment on this.

Response 2: The type of *SECTION_SOLID selected in this paper is ELFORM10, and the numerical simulation results are compared with the experimental group. The results show that the type selected in this paper can accurately simulate the deformation process and force of the lattice structure.

Point 3: In the description of the contact formulation, I would expect also a self-contact condition of both the lattice and the thin-walled body. However, these are omitted in the manuscript whatsoever. Could you provide further explanation?

Response 3: The lattice structure and the self-contact condition "AUTOMATIC_SINGLE_SURFACE" for thin-walled tubes have been added to the manuscript.

Point 4: The coefficients of friction are provided, but without reference to their source.

References to friction coefficients were added to the manuscript.

Point 5: The quasi-static experiments were performed using the load rate of 2 mm/s, while the boundary condition in the numerical simulation is set to 1 m/s, i.e., 500 times higher load rate. The motivation for the selection of the velocity in the numerical simulations should be thoroughly explained, particularly with respect to the lack of experimental verification and the resulting strain rate effects in the lattice structure.

Response 5: To evaluate the energy absorption performance, the tube is usually compressed under quasi-static conditions. In real quasi-static experiments, the loading rate is usually set to 0.01-2 mm/s. However, it is too slow for the explicit finite element algorithm and a lot of time will be spent on calculations. The compression rate for numerical simulations is set to 1000 mm/s to improve the computational efficiency of the finite element model. Literature references have been added to the manuscript. A large number of literature shows that the strain rate effect of the aluminum alloy material can be neglected for compression speeds less than 1000 mm/s. The curve changes in the numerical simulation are almost non-existent and cause very little change in energy absorption.

Point 6: Review the captions of individual figures, where subfigures are present but without context in the caption itself (e.g., Fig. 8).

Response 6: Changes have been made in the manuscript to the image captions throughout the text, adding detailed descriptions of the sub-images.

Point 7: The deformation absorption characteristics of the structures are analyzed with regard to various aspects. However, I am missing cross-referencing of the obtained results with current state-of-the-art in the field, i.e., include a discussion section in your manuscript.

Response 7: The comparison of energy absorption characteristics with the hybrid structure of aluminum foam reinforced thin-walled tube and the discussion were added to the manuscript, and it was found that the hybrid structure designed in this paper improved the SEA by up to 52.5% over the conventional structure.

Point 8: Schematics of the lattice and the thin-walled tube in Figure 1 might be confusing for the readers without cross-referencing throughout the manuscript. I recommend to rotate the visualization of the components around the vertical axis.

Response 8: In the manuscript, Figure 1 is cross-referenced at line 137.

Reviewer 2 Report

The authors considered an energy absorption box consisting of a square tube reinforced by an inner lattice BCC structure. The results of the crashworthiness study are presented in tables and graphically, and are properly analyzed. The paper is well organized and written. There is no doubt that it will be of interest to the readers of the journal Materials. The main question is what will change in the case of dynamic loading that is more realistic for the car's energy absorption box. But it seems that the answer to this question is beyond the manuscript. Because the authors presented a very interesting and important study, including not only experiments and computer simulations but also a detailed analysis of the influence of different elements of the structure and the possible ways of its optimization. So, my conclusion is that the manuscript can be accepted for publication after a minor revision, which is necessary due to a possible misunderstanding of some important points.

My questions and comments that must be addressed are as follows:
1. The authors used three terms for the basic elements of the internal structure: "supporting rod", "pillar", "strut", "cylinder", and "branch". It should be unified or explained directly that these terms are equivalent.
2. Line 172, string "3*3*" should be changed to "3*3*12".
3. Line 229, the figure caption should be changed to "True stress-strain curves of the materials."
4. Table 2, the last column title should be changed to "Ultimate strain".
5. Please cite the corresponding reference in line 303 after "proposed by Santosa et al.", or remove this name.
6. Line 317, remove "is modeled" at the beginning.
7. Lines 312-319, it is not clear what are "tube" and "thin-waled tube" in the description of the contact interactions. Please clarify.
8. Lines 339-340, remove "were used to evaluate the crashworthiness of the structures".
9. Line 354, change "as in Equation" to "as in Equation (3)"
10. Line 359, please put Eq. (4) after this line (after the sentence ending with "with m denoting the total mass of the structure: ")
11. Line 786, change " the compression force is a gradual increasing trend" to " the compression force shows a gradual increasing trend" or something like that.
12. Please double check the use of capital letters at the beginning of sentences.

Author Response

Point 1: The authors used three terms for the basic elements of the internal structure: "supporting rod", "pillar", "strut", "cylinder", and "branch". It should be unified or explained directly that these terms are equivalent.

Response 1: It has been changed to "supporting rod"

Point 2: Line 172, string "3*3*" should be changed to "3*3*12".

Response 2: Made changes in the paper.

Point 3: Line 229, the figure caption should be changed to "True stress-strain curves of the materials."

Response 3: The paper has been revised according to your suggestion.

Point 4: Table 2, the last column title should be changed to "Ultimate strain".

Response 4: Made changes in table2.

Point 5: Please cite the corresponding reference in line 303 after "proposed by Santosa et al.", or remove this name.

Response 5: Removed

Point 6:  Line 317, remove "is modeled" at the beginning.

Response 6: Removed "is modeled" at the beginning.

Point 7: Lines 312-319, it is not clear what are "tube" and "thin-waled tube" in the description of the contact interactions. Please clarify.

Response 7: Uniformly changed to "thin-waled tube" in the paper.

Point 8: Lines 339-340, remove "were used to evaluate the crashworthiness of the structures".

Response 8: Removed in the paper.

Point 9: Line 354, change "as in Equation" to "as in Equation (3)"

Response 9: Changes have been made in the paper.

Point 10:  Line 359, please put Eq. (4) after this line (after the sentence ending with "with m denoting the total mass of the structure: ")

Response 10: Changes have been made as required.

Point 11: Line 786, change " the compression force is a gradual increasing trend" to " the compression force shows a gradual increasing trend" or something like that.

Response 11: Changes have been made as required.

Point 12:  Please double check the use of capital letters at the beginning of sentences.

Response 12: Double-checked the use of capital letters at the beginning of sentences and made changes.

Reviewer 3 Report

Manuscript materials-2232689 entitled “Crashworthiness study of 3D printed lattice reinforced thin-walled tube hybrid structures” is focused on different factors that have influence on absorption of mechanical energy and crashworthiness of 3D printed lattice hybrid structures. The lattice structures were produced with different number of cross-sectional lattice cells and different density distribution (i.e., the effect of angle gradient and strut diameter). The experimental results and mathematical simulations of the hybrid structures are very interesting and are beneficial for practice. The manuscript is quite extensive and is generally well written with well-organized sections. However, I have minor recommendations to improve the quality of this manuscript:

1.     There are quite a lot of formal errors in the manuscript, such as typos, certain language deficiencies, etc., e.g.:

-         lines 179 and 184: Figures 3(e) and 3(f)

-         line 172: 3*3*

-         lines 187-188: Only 7 relative densities?

-         brackets in Table 2

-         line 257: Experimental setup and finite element model Results

-         relatively many minor language deficiencies: form 3 and 4 The folded (line 383), top. the ULT2 (line 387), side The ULT2 (line 388), displacement The final (line 406), determine Se (line 665) etc. I could give many more examples. Therefore, it is necessary to check the English grammar carefully in the whole manuscript.

2.     Figure 2, line 161: The width of the studied thin-walled tube is 30 mm. Is it the total width (i.e., including the tube wall) or without considering the wall thickness (i.e., the inner width of the tube)? It should be better specified.

3.     What is the meaning of the abbreviation Bal. in Table 1? It may not be entirely clear to the reader. Write the full word rather.

4.     Lines 628-631: “For example, the ULT2 hybrid structure with 2 mm tube thickness absorbs 5.32 times more energy, 1.32 times more specific energy absorption, and 5.33 times more load carrying capacity than the hybrid structure with 0.5 mm tube thickness”. The above-mentioned ratios are confusing. For example, for the ULT2 sample (see Table 5), the EA ratio is 1006.56/159.00 = 6.33. Similarly for the other quantities. Explain it. You can also use percentages for the comparison.

5.     Figure 16: The curve ULT2-d is not very clear. It is suitable to use a different color for this curve.

6.     Table 7: The unit of the height of each floor should be specified.

Author Response

Point 1: There are quite a lot of formal errors in the manuscript, such as typos, certain language deficiencies, etc., e.g.:

-         lines 179 and 184: Figures 3(e) and 3(f)

-         line 172: 3*3*

-         lines 187-188: Only 7 relative densities?

-         brackets in Table 2

-         line 257: Experimental setup and finite element model Results

-         relatively many minor language deficiencies: form 3 and 4 The folded (line 383), top. the ULT2 (line 387), side The ULT2 (line 388), displacement The final (line 406), determine Se (line 665) etc. I could give many more examples. Therefore, it is necessary to check the English grammar carefully in the whole manuscript.

Response 1: The above errors have been corrected, and other grammatical and spelling errors in the manuscript have been carefully checked and corrected.

Point 2:  Figure 2, line 161: The width of the studied thin-walled tube is 30 mm. Is it the total width (i.e., including the tube wall) or without considering the wall thickness (i.e., the inner width of the tube)? It should be better specified.

Response 2: The "30 mm" refers to the inner width of the thin-walled tube and is explained in the manuscript.

Point 3: What is the meaning of the abbreviation Bal. in Table 1? It may not be entirely clear to the reader. Write the full word rather.

Response 3: Change "Bal." to "Balance"

Point 4: Lines 628-631: “For example, the ULT2 hybrid structure with 2 mm tube thickness absorbs 5.32 times more energy, 1.32 times more specific energy absorption, and 5.33 times more load carrying capacity than the hybrid structure with 0.5 mm tube thickness”. The above-mentioned ratios are confusing. For example, for the ULT2 sample (see Table 5), the EA ratio is 1006.56/159.00 = 6.33. Similarly for the other quantities. Explain it. You can also use percentages for the comparison.

Response 4: Replace the paragraph with "It can be seen that as the tube wall thickness increases, the peak initial impact force increases significantly, while the energy absorption, specific energy absorption and load-bearing capacity of the structure are greatly enhanced, all due to the more severe plastic deformation caused by the larger mass of the structure. For example, the ULT2 hybrid structure with a tube thickness of 2 mm is 632% of the energy absorbed, 232% of the specific energy absorption, and 633% of the load carrying capacity of a hybrid structure with a tube thickness of 0.5 mm. In addition, CLE also shows better results at larger thicknesses and increases with the number of transverse cells."

Point 5: Figure 16: The curve ULT2-d is not very clear. It is suitable to use a different color for this curve.

Response 5: Changes were made to the color of ULT2-d in Figure 16 in the manuscript to make it clear.

Point 6: Table 7: The unit of the height of each floor should be specified.

Response 6: The height unit "mm" for each floor unit has been added to Table 7.